# SO$_2$ and copper tolerance exhibit an evolutionary trade-off in *Saccharomyces cerevisiae*

**Cristobal A. Onetto**[1], **Dariusz R. Kutyna**[1], **Radka Kolouchova**[1], **Jane McCarthy**[1], **Anthony R. Borneman**[1,2], **Simon A. Schmidt**[1] *

**1** The Australian Wine Research Institute, Glen Osmond, South Australia, Australia, **2** School of Agriculture, Food and Wine, Faculty of Sciences, University of Adelaide, Adelaide, South Australia, Australia

* simon.schmidt@awri.com.au

**Data Availability Statement:** All relevant data are within the manuscript or have been deposited with the following specialist or general data repositories. The genome sequence of strains AWRI 3811,

## Abstract

Copper tolerance and SO$_2$ tolerance are two well-studied phenotypic traits of *Saccharomyces cerevisiae*. The genetic bases of these traits are the allelic expansion at the *CUP1* locus and reciprocal translocation at the *SSU1* locus, respectively. Previous work identified a negative association between SO$_2$ and copper tolerance in *S. cerevisiae* wine yeasts. Here we probe the relationship between SO$_2$ and copper tolerance and show that an increase in *CUP1* copy number does not always impart copper tolerance in *S. cerevisiae* wine yeast. Bulk-segregant QTL analysis was used to identify variance at *SSU1* as a causative factor in copper sensitivity, which was verified by reciprocal hemizygosity analysis in a strain carrying 20 copies of *CUP1*. Transcriptional and proteomic analysis demonstrated that *SSU1* over-expression did not suppress *CUP1* transcription or constrain protein production and provided evidence that *SSU1* over-expression induced sulfur limitation during exposure to copper. Finally, an *SSU1* over-expressing strain exhibited increased sensitivity to moderately elevated copper concentrations in sulfur-limited medium, demonstrating that *SSU1* over-expression burdens the sulfate assimilation pathway. Over-expression of *MET 3/14/16*, genes upstream of H$_2$S production in the sulfate assimilation pathway increased the production of SO$_2$ and H$_2$S but did not improve copper sensitivity in an *SSU1* over-expressing background. We conclude that copper and SO$_2$ tolerance are conditional traits in *S. cerevisiae* and provide evidence of the metabolic basis for their mutual exclusivity. These findings suggest an evolutionary driver for the extreme amplification of *CUP1* observed in some yeasts.

## Author summary

Completing a commercial wine fermentation is a tough job for a yeast. Grape juice is a highly variable environment and to cope with that variability, a large number of different yeast strains have been generated exhibiting different features. Two of the most distinguishing physical characteristics of wine yeast are copper and SO$_2$ tolerance, which appear to be mutually exclusive. The genetic underpinnings of these two traits are individually

AWRI 3471, AWRI 3807, AWRI 3001 and AWRI 3470 used in the analysis of QTL following bulk segregant sequence analysis and the raw RNA-seq reads are available in NCBI under Bioproject PRJNA877711. Quantitative proteomic data have been made available via the Proteomics Identifications Database (PRIDE) at http://www.ebi.ac.uk/pride/archive/projects/PXD037997. Raw data is available at Dryad at https://doi.org/10.5061/dryad.wdbrv15s5.

**Funding:** This work was supported by a grant from Wine Australia (AWRI 1701-3.2.2 to SA), with levies from Australia's grapegrowers and winemakers and matching funds from the Australian Government. The funders had no role in study design, data collection and analysis, decision to publish, or preparation of the manuscript.

**Competing interests:** The authors certify that they have no affiliations with or involvement in any organisation or entity with any financial interest (such as honoraria; educational grants; participation in speakers' bureaus; membership, employment, consultancies, stock ownership, or other equity interest; and expert testimony or patent-licensing arrangements), or non-financial interest (such as personal or professional relationships, affiliations, knowledge or beliefs) in the subject matter or materials discussed in this manuscript.

well-characterised, but there doesn't appear to be an obvious reason why copper tolerance and SO$_2$ tolerance could not co-exist. We performed a genetic analysis that showed how over-expression of the SO$_2$ transporters responsible for SO$_2$ tolerance induced copper sensitivity. Our analysis of RNA and protein levels in SO$_2$-tolerant yeast showed that they could still produce the molecules that would usually protect them when exposed to copper stress. However, the constant activation of the transporter that provides SO$_2$ tolerance also induced a sulfur limitation that could not be overcome when combined with copper stress.

## Introduction

Copper and SO$_2$ are used nearly ubiquitously in the wine industry. Their usefulness is, at least in part, due to their varied activities. In the vineyard, copper- and sulfur-based sprays are applied to control both downy [1] and powdery mildews [2]. In the form of SO$_2$, sulfur is used during grape processing to help protect harvested grapes, juice and must against unwanted microbial activity [3] and oxidation [4]. Likewise, it is used after fermentation to stabilise the finished wine. Copper is used in finished wine to moderate aromas derived from low molecular weight sulfur compounds [5]. As a result of this multitude of applications, copper and sulfur have been part of the grape grower and winemaker tool kit for generations.

The commercial and environmental prevalence of copper and SO$_2$ has provided an environment where resistance to these two compounds has manifested in wine yeast and environmental isolates, including medical specimens [6]. Freshly prepared grape juice typically contains between 0.5 and 1.5 mg/L of copper, although concentrations above 7 mg/L [7,8] can be observed, depending on copper usage in the vineyard [9].

The activities of copper in yeast are diverse and complex, as is its regulation [reviewed in 10]. Copper resistance is mediated predominantly by the metallothionein Cup1p [11] and, to a lesser extent, Crs5p [12], Sod1p [13] and glutathione [14]. Copy number variation of the *CUP1* gene is commonly observed in *S. cerevisiae* and has been estimated at between 0–70 copies per cell [6,11,15–17], with higher copy numbers being associated with higher levels of resistance to otherwise inhibitory concentrations of copper [18,19]. Despite the primary association of *CUP1* with copper tolerance, copy number expansion explains only 44.5% of phenotypic variation in copper tolerance [20]. Other mechanisms by which *S. cerevisiae* responds to copper excess include Mac1p dependent control of copper import [21], manipulation of the copper oxidation state via Fet3p [22] and *SLF1*-mediated mineralisation of copper into copper sulfide [23].

SO$_2$ induces its deleterious effects on yeast by compromising energy metabolism. Inhibition of glycolysis decreases ATP production, while membrane leakage that results from membrane damage increases its consumption [reviewed in 24]. SO$_2$ tolerance in *S. cerevisiae* is mediated by the efflux pump Ssu1p [25]. Unlike the mechanism that gives rise to copper tolerance, variation in SO$_2$ tolerance in wine strains of *S. cerevisiae* is predominantly the result of reciprocal translocations between chromosomes VIII and XVI [26,27] although translocations between chromosomes XV and XVI [28] and an inversion within chromosome XVI [29] have also been identified in a limited number of isolates. *SSU1* expression in wild type *S. cerevisiae* is regulated by Fzf1p [30]. As a result of its fusion with the promoter of *ECM34*, control of *SSU1* expression is freed from Fzf1p regulation in strains carrying the VIII::XVI translocation [31]. Substantial heterogeneity exists in the precise structure of the *ECM34* promoter with variable expression of *SSU1* and subsequent variance in SO$_2$ tolerance as a result [31,32].

The sulfate assimilation pathway has also been implicated in resistance to SO$_2$ in a coordinated response mediated by *COM2* [33]. It is suggested that components of this pathway contribute to SO$_2$ resistance by reducing SO$_2$ to H$_2$S which can then either exit the cell or further metabolised by condensation with o-acetyl homoserine [for review see 34].

By constitutively exporting SO$_2$, over-expression of *SSU1* directly intervenes in the sulfate assimilation pathway, acting to remove the immediate precursor to hydrogen sulfide [reviewed in 34]. A dependency on sulfur metabolism genes as a response to extremes of copper stress has previously been noted [35] as has a relationship between SO$_2$ tolerance and copper sensitivity in wine yeast [36].

Hodgins-Davis et al. [35] show that both copper starvation and toxicity elicit responses from genes associated with sulfur metabolism. *CUP1* poor strains were shown to respond to increased copper concentrations with upregulation of genes related to mitochondrial activity or oxidative stress. These cellular functions are also critical contributors to survival in multiple forms of starvation [37] and are part of a larger response to starvation which ultimately results in cell cycle arrest [38].

In other work, Fay et al [39] observed upregulation of sulfate assimilation pathway genes in a subset of strains during a study of copper stress but could not directly associate this as a response to copper, but rather with the capacity of those strains to become discolored through the formation of CuS. de Freitas et al [40] showed that addition of copper could rescue the ability of Δssu1 yeast to grow on non-fermentable carbon sources. This was attributed to depletion of copper or iron by accumulation of intracellular sulfur compounds. Taken together these works provide a picture of the strong interdependence of copper and sulfur metabolism in yeast.

To determine the contribution of *CUP1-2* copy number variation to copper tolerance in wine yeast, we determined the copy number status of 94 wine yeast strains and compared copy number variation to competitive fitness scores in a copper-containing medium. The comparison identified many strains that exhibited copper sensitivity despite carrying a high number of *CUP1-2* copies. The genetic contributions to copper sensitivity in selected strains were determined by mating strains with equivalent or differential *CUP1-2* copy numbers and/or fitness in high copper media, followed by bulk segregant analysis of progeny. Potential genetic contributions to copper sensitivity were confirmed through reciprocal deletions and over-expression analysis in parental lines. Transcriptomic and proteomic analysis identified a potential metabolic limitation induced by growth in high copper in SO$_2$ tolerant wine yeast. The degree to which SO$_2$ tolerance contributed to the metabolic limitation was evaluated using a series of fermentation trials.

## Results and discussion

### *CUP1* copy number alone is a poor predictor of copper tolerance in wine yeast

The *CUP1* copy number in 94 wine yeast strains, normalised against a single insert molecular barcode, was determined using qPCR. *CUP1* copy number varied dramatically between 2 (SD 0.1) and 55 (SD 2.8) absolute copies per cell [41, file T04]. This is consistent with the 0–26 haplotype copies (*CUP1-1* + *CUP1-2*) for strains in the wine yeast clade (Fig B in S1 Text) estimated from whole-genome sequence data [15].

With the previously observed diversity in wine yeast copper tolerance [36] and the high diversity in *CUP1* copy number among these strains we expected a strong relationship between copy number and fitness. However, no correlation between *CUP1* copy number and tolerance to copper was observed (Fig 1). Many strains exhibited both a significant positive fitness

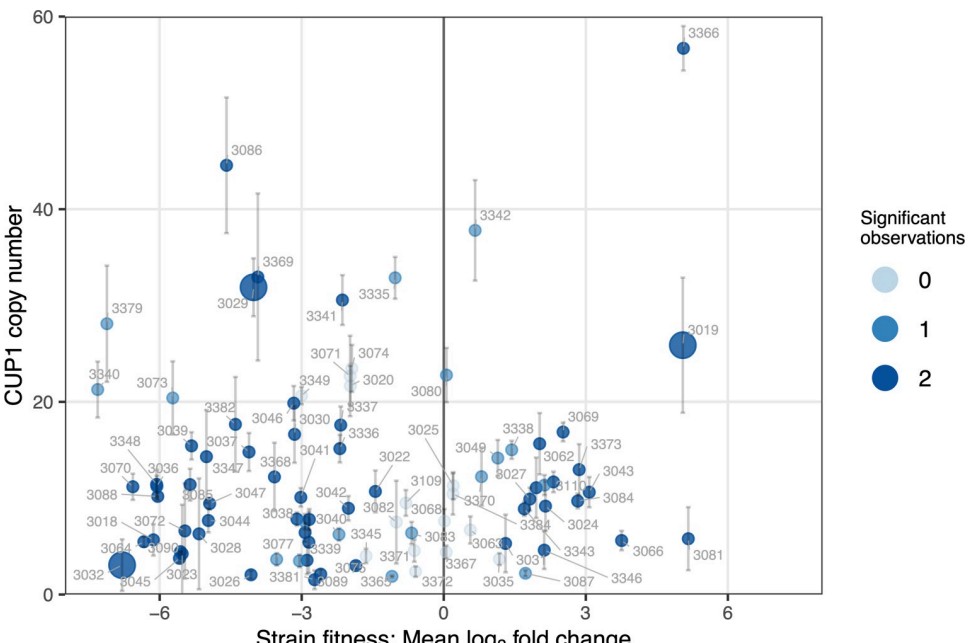

**Fig 1. Relationship between yeast strain fitness in high copper medium and *CUP1* copy number.** The fitness value shown on the x-axis is the mean of two independent experiments. Significant observations are recorded as 0, 1 or 2 if there was evidence ($P < 0.05$) that the log2 fold change differed relative to the control condition in independent fitness experiments (n = 3 for each of 2 independent experiments). The y-axis shows the mean absolute *CUP1* copy number of each strain. Error bars show standard deviation (n = 3). Strains pictured with a larger point size (3019, 3032 and 3029) were the parental strains used in subsequent bulk segregant QTL experiments.

attribute and high *CUP1* copy number (between 6 and 18 copies). However, an equally large number of strains with a high *CUP1* copy number exhibited poor fitness in 10 mg/L of copper.

## QTL analysis of segregants identifies *SSU1* as a contributing factor in copper sensitivity

The observation here and elsewhere [20] that *CUP1* copy number is a poor predictor of copper tolerance in yeast raises the obvious question; why are strains with a high *CUP1* copy number so poorly tolerant of copper in the medium? The question of copper sensitivity among *CUP1* containing yeast was addressed using bulk segregant analyses of haploid strains derived from crosses between a single copper tolerant parent with high *CUP1* copy number (AWRI 796, 25.6 [SD 5.3] copies) and two copper sensitive parents, with either low (AWRI 1537, 2.5 [SD 0.9]) or high (AWRI 1487, 31.7 [SD 2.7]) copies of *CUP1*, respectively (highlighted with large spots in Fig 1).

Stable haploid progeny were obtained for each of the parental genotypes, through the prior inclusion of a molecular barcode that disrupted *HO* [described in 36]. The parental copper-resistance phenotype was present in all spores isolated from AWRI 3019, AWRI 3029 and AWRI 3032 (Fig C in S1 Text) corresponding to barcoded versions of AWRI 796, AWRI 1537 and AWRI 1487, respectively. It should be noted that the copper tolerance phenotype was only assessed in a subset of the isolated spores (i.e. those containing a barcode and containing complementary mating type genes).

A copper-tolerant haploid derived from AWRI 3019 (Cu$^{tol}$: CUP1$^{high}$) was mated with copper sensitive haploids derived from AWRI 3029 (Cu$^{sen}$: CUP1$^{low}$) and AWRI 3032 (Cu$^{sen}$: CUP1$^{high}$) to yield two diploid strains, AWRI 3001 and AWRI 3811. In each case, the diploid

derivatives were copper sensitive, indicating that this is the dominant phenotype. To map F1 phenotypes, both AWRI 3001 and AWRI 3811 were sporulated. Tetrad dissection of spores routinely recovered 3 to 4 viable spores. In total, 80 and 146 spores were isolated and phenotyped for copper sensitivity for the two crosses, which were subsequently divided into either copper-tolerant (n = 41 and 67) or copper-sensitive (n = 32 and 51) pools (excluding a small number of spores with an ill-defined phenotype) (Fig A in S1 Text).

An analysis of the SNP frequency in F1 progeny from AWRI 3001 (Cu$^{tol}$:CUP1$^{high}$ x Cu$^{sen}$: CUP1$^{low}$) was undertaken. One hundred and one positions across the genome were identified that exhibited a SNP ratio greater than 0.85. Sixty-five of those were on Chr VIII with a peak between 210,000 and 218,000 bp. This position corresponds to the location of *CUP1-1* and *CUP1-2* (Fig 2A). The major association on Chr VIII was consistent with the mean difference in *CUP1* copy number between the two strains (Δ copy number = 23.1 copies [95CI, 19.9, 26.3]). This data explains the copper sensitivity of the AWRI 3029 strain and supports previous observations [11,12,17,18] that *CUP1* is a key determinant of copper tolerance in yeast.

Of the remaining genomic locations with SNP ratios greater than 0.85, eleven of them were on Chr IV between positions 1,163,450 and 1,163,515 bp. However, the limited breadth of the change in SNP frequency around this location and that this, and other locations with high SNP ratios, were not mirrored in Cu-tol and Cu-sens pools suggest that there is no association with the phenotype.

QTL analysis in the progeny of the Cu$^{tol}$: CUP1$^{high}$ x Cu$^{sen}$: CUP1$^{high}$ cross (AWRI 3811) showed a divergence in SNP frequencies approaching 100% (for the sensitive and resistant parental genotypes in the copper sensitive and resistant pools) on the extreme left arm of Chr VIII and between 350,000 bp and 400,000 bp on Chr XVI (Fig 2B). There were few other genomic locations where either parental SNP frequency exceeded 0.75 in this data set. The two positions of the QTLs on Chr VIII and XVI are consistent with the position of the previously described translocation between the genes *SSU1* and *ECM34* [26]. It is noteworthy that the two parents of this cross have either wild-type (AWRI 3019) or translocated (AWRI 3032) chromosomes at the *SSU1* locus [41, file T05]. Translocations at this position have previously been associated with increased SO$_2$ tolerance in yeast [26] but associations with copper sensitivity have not been reported.

A divergence from the expected 0.5 SNP ratios for the entirety of Chr I was observed in diploids derived from both crosses (AWRI 3001 and AWRI 3811) and copper tolerant and copper sensitive pools prepared from the respective F1 progeny (Fig 2A and B). The divergence from expected SNP ratios for Chr I can be explained by a whole chromosome duplication that has been reported previously [42] in the progenitor strain used for this work (AWRI 796).

## Deletion of *SSU1* restores copper tolerance in copper sensitive hybrid

The contribution of *SSU1* to copper sensitivity was evaluated by reciprocal deletion of *SSU1* in the copper-sensitive hybrid AWRI 3811 (Cu$^{tol}$: CUP1$^{high}$: *SSU1*$^{WT}$ x Cu$^{sen}$: CUP1$^{high}$: *SSU1*-$^{trans}$) to generate AWRI 3901 (*SSU1*$^{WT/\Delta}$) and AWRI 3902 (*SSU1*$^{\Delta/trans}$). The growth of each of these strains, in addition to the haploid parents of AWRI 3811, was evaluated in a defined medium with copper concentrations of 0.25 mg/L or 10 mg/L (Fig 3).

While deleting the wild type copy of *SSU1* from Chr VIII had no effect on the sensitivity of AWRI 3811, deleting *SSU1* from the translocated VIII::XVI chromosome restored copper tolerance to the hybrid (Fig 3). This demonstrates that *SSU1* on the translocated chromosome is the causative factor of copper sensitivity in the AWRI 3811 hybrid.

The negative effect of *SSU1* on copper tolerance is entirely due to its level of expression. This is demonstrated in Fig 4A which compares the growth of the copper-tolerant haploid

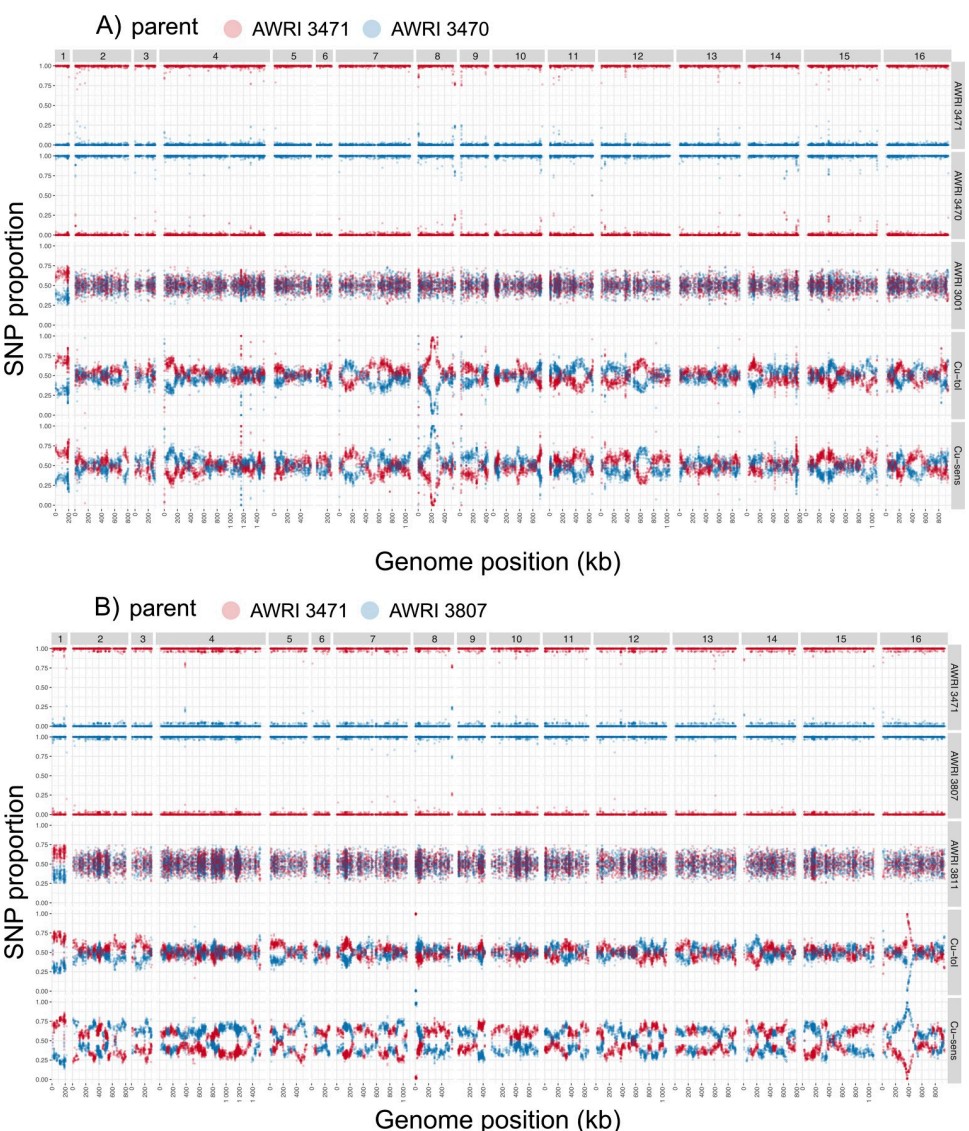

**Fig 2.** Single nucleotide variant (SNV) frequency in spores generated from diploids following crosses between A) AWRI 3471 and AWRI 3470, and B) AWRI 3471 and AWRI 3807. SNV frequency is shown for each parent, the diploid generated from each cross (AWRI 3001 and AWRI 3811) and spore pools where each spore in the pool was classified as either copper tolerant (Cu-tol) or copper sensitive (Cu-sens).

AWRI 3471 with AWRI 4052 (*ssu1(pr)Δ::ECM34(pr)*), a strain in which the wild-type promoter of *SSU1* in the AWRI 3471 background was exchanged for the *ECM34* promoter from AWRI 1487. These near isogenic strains were compared in a defined medium containing 0.25 mg/L and 10 mg/L of copper. As in Fig 3, AWRI 3471 does not show any growth inhibition in the presence of elevated copper concentrations. Except for the promoter of *SSU1*, AWRI 4052 is genetically identical to AWRI 3471. However, AWRI 4052 is copper sensitive, exhibiting a mean biomass decrease of 2.1 g/L DCW [95CI, 1.8, 2.4].

Fig 4B compares the fermentation performance of AWRI 3471 and AWRI 4052. In addition to the effects on growth, elevated copper concentrations in grape juice have been shown to impede sugar utilisation [43,44], an effect that is particularly relevant to commercial winemaking. Elevated copper concentrations effected fermentation progress in both AWRI 3471 and

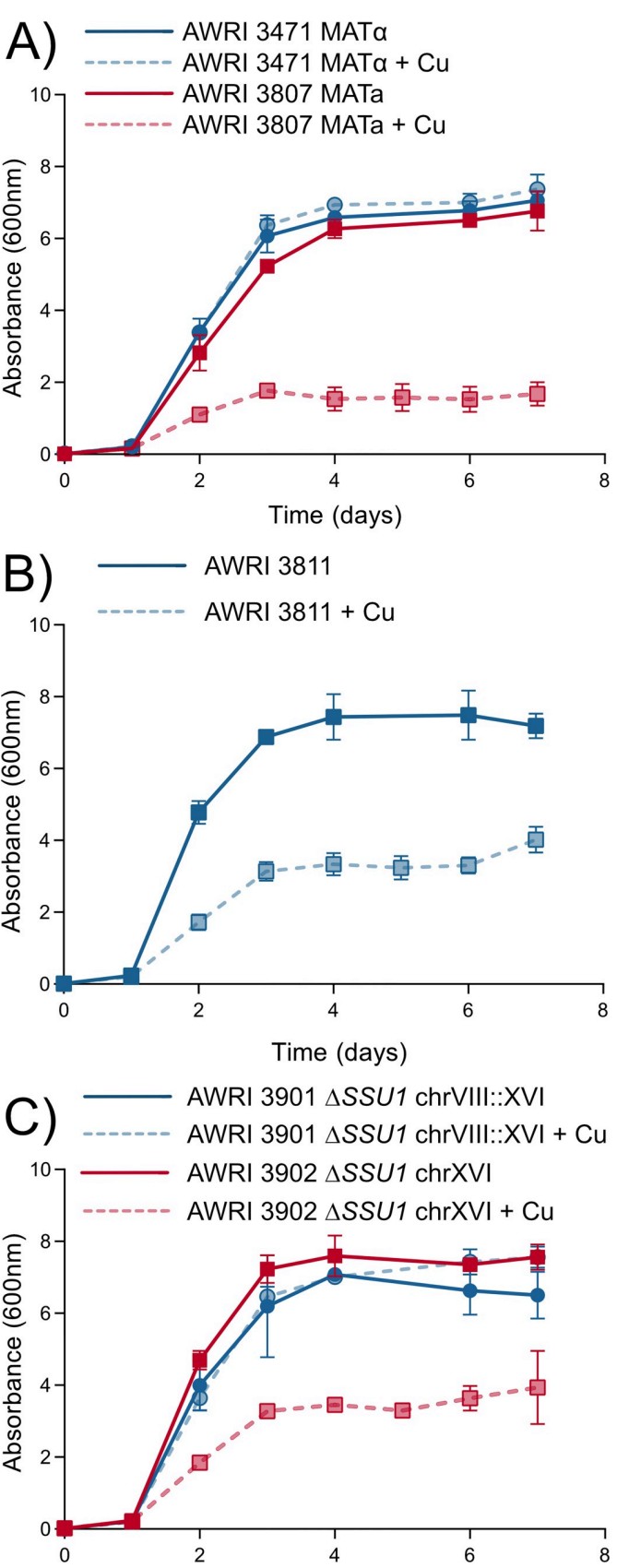

**Fig 3. Heritability of *SSU1* dependent copper tolerance and sensitivity assessed in defined medium containing either 0.25 or 10 mg/L copper.** (A) Growth of yeast AWRI 3471 (●) and AWRI 3807 (□), haploid derivatives of AWRI 796 and AWRI 1487 respectively. (B) Growth of AWRI 3811, a diploid derived from a cross between 3471 x 3807. (C) Growth of AWRI 3901 and AWRI 3902, two derivatives of the AWRI 3811 diploid each containing a deletion of *SSU1* at chromosome XVI and VIII::XVI respectively. Filled lines; standard defined medium, dashed lines; defined medium containing 10 mg/L copper. Points show mean of three replicates with error bars indicating standard deviation.

AWRI 4052, however, a severe delay in fermentation onset and significantly higher residual sugar concentrations were observed in fermentations by strain AWRI 4052 in copper excess relative to strain AWRI 3471 (mean increase of 78 g/L [95CI, 67, 86]) at day 17.

## The combined effect of *SSU1* over-expression and high copper concentration on yeast gene expression during fermentation

The effect of *SSU1* over-expression on copper sensitivity was explored through an analysis of the transcriptional response of strains bearing *SSU1* and *ssu1(pr)Δ::ECM34(pr)* grown in a medium containing 10 mg/L of copper [41, file T10]. Genes for which there was strong evidence of differential abundance (P < 0.005) and for which the magnitude of the change was greater than 2-fold (Log$_2$FC > 1) were further subjected to over-representation analysis, using Gene Ontology (GO) Biological Process as a grouping category. Fig 4C shows the relative expression as Log$_2$FC (AWRI 4052—AWRI 3471) of the filtered gene set, grouped according to the GO category with which they are associated.

Notably, the two strains cannot be differentiated according to the Cup2p or Msn2p responsive genes. *CUP1* transcript abundance, for example, is equivalent between the two genotypes.

The most prominent distinguishing transcriptional features are related to sulfur compound transport (Fig D in S1 Text, enrichment log$_{10}$(P) = -7.33). Specifically, increased expression of genes associated with sulfate uptake (*SUL1*, *SOA1*), sulfonate catabolism (*JLP1*) and sulfur-containing amino acid uptake (*MUP1*, *MUP3*, *YCT1*, *OPT1*, *AGP3*) in AWRI 4052 relative to AWRI 3471 indicate that over-expression of *SSU1* increases the burden on sulfur metabolism beyond that imposed by copper alone. Furthermore, the genes *AGP3*, *PDC6* and *YRO2*, which have previously been postulated to be markers of sulfur-limited growth [45], are all up-regulated in this contrast.

The general down-regulation of cell wall structural components under copper stress is accentuated in cells over-expressing *SSU1* with greater down-regulation of mannoproteins (*TIR1*, *TIR2*, *TIR3*, *TIR4*, *DAN1*), and seripauperins (*PAU17*, *PAU5*, *PAU16*).

A feature of the expression profile of AWRI 4052 grown under high copper is the down-regulation of genes related to thiamine metabolism. Not only is the expression of *THI4* decreased (Log$_2$FC = -1.1, P = 0.006), but a down-regulation of the broader thiamine regulon is evident. Down-regulation of *THI7* (Log$_2$FC = -1.2), *THI73* (Log$_2$FC = -1.1), *THI20* (Log$_2$FC = -1.6) and *PDC5* (Log$_2$FC = -2.4) in the *ssu1(pr)Δ::pECM34(pr)* background suggest that these cells are sensing excess thiamine [46]. Thiamine accumulation may be a consequence of a slowing growth rate in yeast over-expressing *SSU1* and experiencing copper stress.

In summary, the copper sensitivity exhibited by the *ssu1(pr)Δ::pECM34(pr)* strain AWRI 4052 cannot be explained by mis-regulation of genes known to be critical in the maintenance of copper homeostasis, such as *CUP1*. The up-regulation of genes associated with either the direct import of sulfur (as sulfate or as sulfur-containing amino acids) or the scavenging of sulfur (from intra-cellular sulfonates) indicates that *SSU1* over-expression exacerbates copper stress by inducing a sulfur limitation. A diagrammatic representation of the sulfate

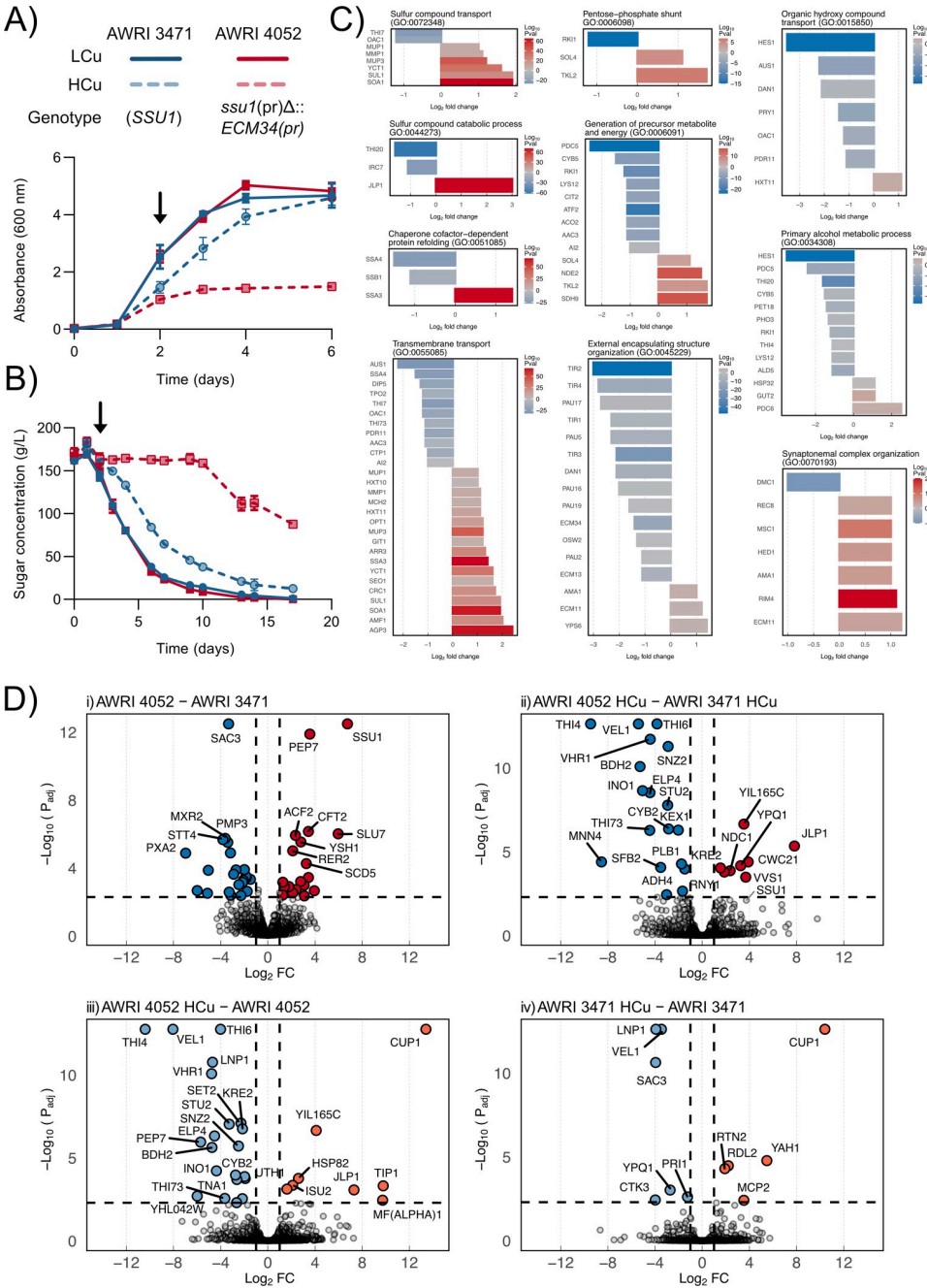

**Fig 4. Effect of copper on fermentation, gene expression and protein production in AWRI 3471 and AWRI 4052.**
(A) Growth as indicated by absorbance at 600 nm and (B) Sugar consumption of *SSU1* wild type (AWRI 3471, blue) and *ssu1*(pr)Δ::ECM34(pr) (AWRI 4052, red) yeast strains during growth in defined medium with and without copper at 10 mg/L. Filled lines; standard defined medium, dashed lines; defined medium containing 10 mg/L copper. Arrow in (A) and (B) show sampling points used for RNAseq analysis. Points show mean of three replicates with error bars indicating standard deviation. (C) Relative transcript abundance grouped by Gene Ontology summary category for the contrast AWRI 4052 HCu–AWRI 3471 HCu, expressed as Log₂ Fold Change (Log₂FC) with red showing increased and blue decreased expression. Colour intensity highlights the P value score obtained from differential expression analysis undertaken with DEseq2. *SSU1* was omitted from over-representation analysis and therefore does not appear in enriched ontology summaries. (D) Relative protein abundance shown as volcano plots with colours indicating increased (red) and decreased (blue) relative fold change for four contrasts i) AWRI 4052 –AWRI 3471, ii) AWRI 4052 HCu–AWRI 3471 HCu. Colours indicating increased (orange) and decreased (light blue) are used in plots iii) AWRI 4052 HCu–AWRI 4052, iv) AWRI 3471 HCu–AWRI 3471. Vertical dotted lines indicate Log₂FC = 1 and horizontal dotted lines indicate an adjusted P value = 0.005. n = 3 in all cases. *SSU1* is shown despite having P value > 0.005 in panel ii).

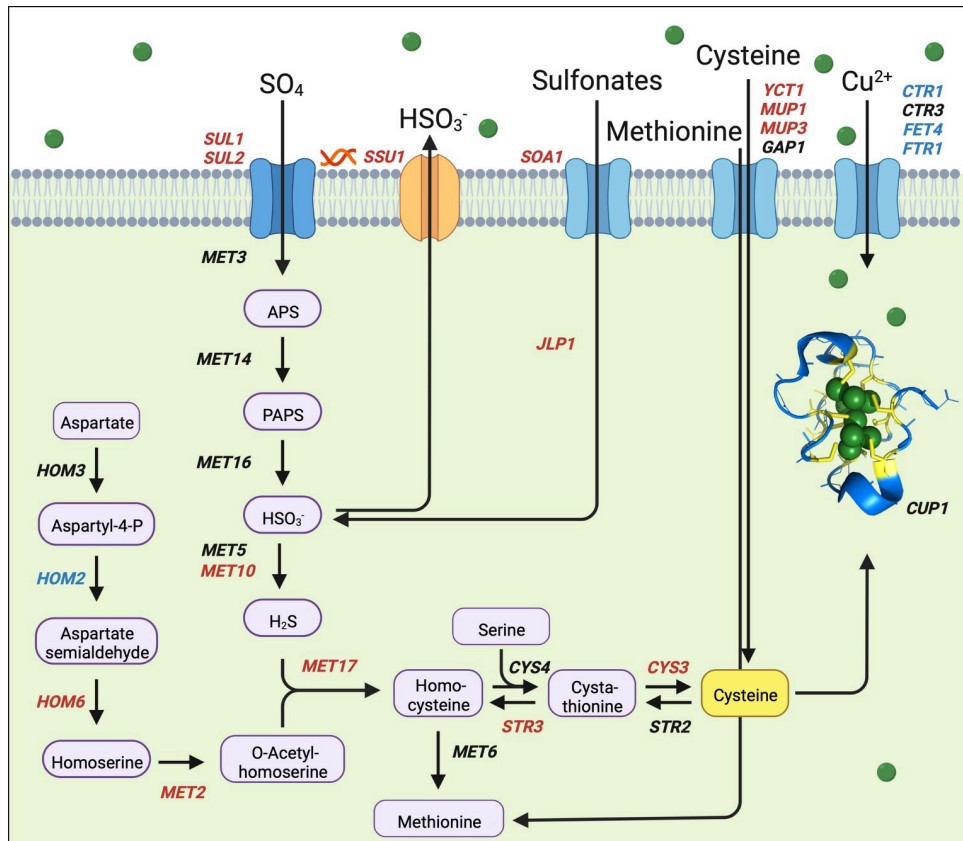

**Fig 5. Diagrammatic representation of the sulfate assimilation pathway.** Gene names in red and blue indicate up-regulated and down-regulated genes, respectively, comparing *SSU1* over-expressing (AWRI 4052) yeast with control yeast (AWRI 3471) growing in medium containing 10 mg/L copper. *SSU1* over-expression is indicated by a red helix. Gene names in black indicate no change in expression. Green circles represent copper ions. The structure of Cup1p is adapted from the crystal structure determined by Calderone et al [48]. Cysteine residues in the structure are coloured yellow. APS; Adenosine-5'-phosphosulfate, PAPS; phosphoadenosine phosphosulfate. The image was created with BioRender.com.

assimilation pathway showing the effect of copper on the expression of genes in the pathway is provided in Fig 5.

*SSU1* encodes an SO$_2$ efflux pump (Ssu1p). SO$_2$ is an intermediate in the sulfate assimilation pathway that is required for the biosynthesis of cysteine and methionine [47]. It is possible that increased activity of the Ssu1p transporter may limit the flux through the sulfate assimilation pathway. If this were the case, sulfur limitation induced by *SSU1* over-expression may contribute to copper sensitivity in a number of ways. Although *CUP1* transcripts are not mis-regulated in *SSU1* over-expressing cells, as demonstrated above, the 53 amino acid product of *CUP1* contains 12 cysteines [48] and therefore sulfur limitation may place a constraint on Cup1p production. Glutathione, a metabolite critical in the response to copper stress [49,50] may be similarly constrained. Alternatively, a restriction of flux through the sulfate assimilation pathway may also constrain the production of hydrogen sulfide, another intermediate in the sulfate assimilation pathway. Hydrogen sulfide has been shown to moderate the toxicity of copper through *SLF1*-mediated CuS mineralisation [23]. In the following sections we will evaluate whether *SSU1* mediated constraints on the sulfate assimilation pathway contribute to copper sensitivity in *S. cerevisiae* through the alternative possibilities discussed above, beginning with the production of Cup1p protein.

## Label-free proteomic analysis demonstrates equivalent Cup1p production in *S. cerevisiae* containing either *SSU1(pr)* or *ssu1(pr)Δ::ECM34(pr)*

To determine whether *SSU1* over-expression inhibits Cup1p formation in high copper medium, a label free quantitative proteomic analysis was undertaken. A total of 3261 proteins were identified in each of the extracts of the same samples used in the analysis of gene expression, collected two days following inoculation (Fig 4D). The *ssu1(pr)Δ::ECM34(pr)* mutation in AWRI 4052 resulted in a 107-fold increase ($P_{adj}$ = 3.23e$^{-13}$) in the abundance of Ssu1p relative to AWRI 3471 when grown in a medium containing standard copper concentrations [41, file T11]. The increase in Ssu1p abundance highlights the effectiveness of the *ECM34* promoter in driving *SSU1* expression. Differential abundance of an additional 77 proteins was observed relating to over-expression of *SSU1* alone.

The contrast in protein abundance between AWRI 4052 and AWRI 3471 in high copper medium identified the following pathways as being over-represented with differentially abundant proteins; primary alcohol biosynthesis, glycoprotein biosynthesis, apoptotic process, polysaccharide metabolic process and energy derived by oxidation of organic compounds (Fig E in S1 Text). There was limited overlap between transcriptomic and proteomic profiles. Six genes/proteins, excluding *SSU1*/Ssu1p, were common to both data sets (*JLP1*, *SSA4*, *MSC1*, *AAC3*, *THI20*, *THI73*). There was a 13-fold increase in Ssu1p abundance ($P_{adj}$ = 0.008), which is a decrease from that observed in a low copper medium [41, file T12].

There was strong evidence for an increase in the abundance of Jlp1p (log$_2$FC = 7.8, $P_{adj}$ = 4.5e$^{-6}$), supporting the idea that sulfur limitation is exacerbated in AWRI 4052 exposed to copper stress. However, there was no evidence for the differential abundance of other identifiers of sulfur-limited growth (Agp3p, Pdcp, Yro2p and Soa1p). A general decrease in the abundance of proteins involved in thiamine biosynthesis (Thi4p, Thi6p, Thi20p, and Thi73p) or thiamine precursor scavenging (Snz2p) was also observed, which is consistent with the gene expression profile of this strain under copper stress.

As expected, the protein with the largest change in abundance in response to high-copper concentrations (high–low copper contrast) was Cup1p with a 10.4 ($P_{adj}$ = 2.4e$^{-13}$) and 13.4 ($P_{adj}$ = 1.7e$^{-13}$) Log$_2$FC in AWRI 3471 and AWRI 4052, respectively. The 6.5-fold relative increase in Cup1p abundance ($P_{adj}$ = 0.02) in the *ssu1(pr)Δ::pECM34(pr)* background demonstrates that inability to produce sufficient metallothionein is not an explanation for copper sensitivity in this strain, but does suggest that increased copper stress is being perceived. There was no evidence ($P_{adj}$ > 0.5) for the differential abundance of proteins whose transcripts had previously been shown to be copper responsive (Oye3p, Fet3p, Ftr1p, Gto3p, Hsp12p, Fet4p and Sod1p).

An interesting feature of the AWRI 4052 'high copper'- 'low copper' contrast was the apparent increase in abundance of MF(alpha)1 protein (Log$_2$FC = 9.7, P = 0.004) despite strong evidence for a decrease in the abundance of its transcript (Log$_2$FC = -2.8, P = 2.8e$^{-268}$). MF(alpha)1 has previously been shown to be a copper-binding protein [51,52] but we cannot explain its increased abundance in *SSU1* over-expressing cells growing in high copper concentrations.

In summary, an examination of relative protein abundance data supports the idea that *SSU1* over-expression exacerbates sulfur limitation in copper challenged yeast and rules out Cup1p limitation as a cause of copper sensitivity. It leaves open the question about the role of thiamine biosynthetic and uptake functions in copper sensitivity.

## *SSU1* over-expression does not limit hydrogen sulfide production

The contribution of H$_2$S metabolism to copper sensitivity was assessed by plasmid-based over-expression of *MET3*, *MET14* and *MET16* ([MET$^+$]) in the two strains AWRI 3471 and AWRI

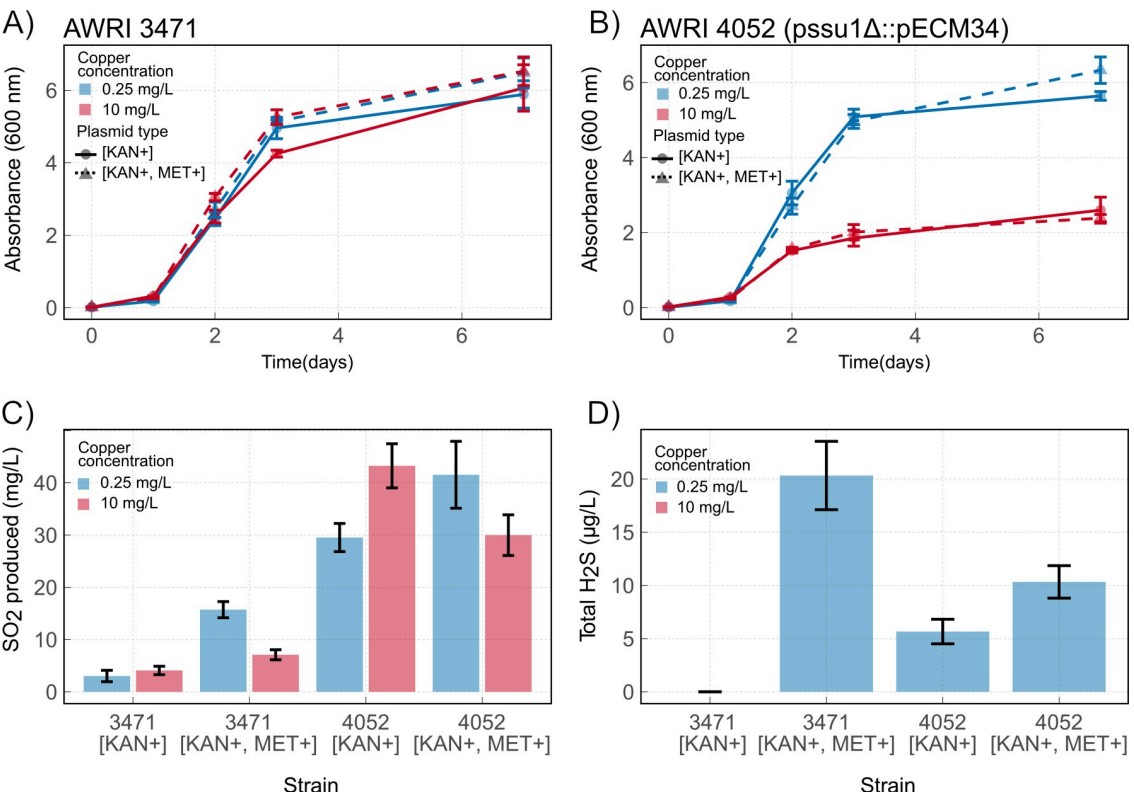

**Fig 6. Effect of copper concentration and MET3-MET14-MET16 over-expression on growth and, SO$_2$ and H$_2$S production in AWRI 3471 and AWRI 4052.** (A) Growth of AWRI 3471 in medium containing 0.25 mg/L and 10 mg/L copper. (B) Growth of AWRI 4052 in medium containing 0.25 mg/L and 10 mg/L copper. (C) The concentration of SO$_2$ produced by AWRI 3471 and AWRI 4052 in medium containing 0.25 mg/L and 10 mg/L copper. (D) The concentration of H$_2$S produced by AWRI 3471 and AWRI 4052 in medium containing 0.25 mg/L copper. Blue lines and bars; standard defined medium, red lines and bars; defined medium containing 10 mg/L copper, filled lines; plasmid carrying KANMX marker only, dashed lines; plasmid carrying KANMX, MET3, MET14 and MET16. Points and bars show mean of three replicates with error bars indicating standard deviation.

4052. It was reasoned that if H$_2$S was limited due to a decrease in the concentration of its precursor, then an increase in flux through the pathway should rectify this condition and restore copper tolerance.

Growth in high and low copper medium was unaltered in either genetic background by increased expression of *MET3*, *MET14* and *MET16* with no alleviation of copper sensitivity evident in AWRI 4052 (Fig 6A and 6B).

Three-way analysis of variance indicated that yeast strain accounted for most of the variance in SO$_2$ production (P < 0.0001) (Fig 6C), an observation that is explained by *SSU1* over-expression in AWRI 4052. Therefore, the effect of copper and MET 3/14/16 expression was analysed separately by strain using two-way ANOVA (Table C and Table D in S1 Text). There was strong evidence for a MET 3/14/16 dependent increase in SO$_2$ production (Fig 6C) in both AWRI 3471 (P < 0.0001) and AWRI 4052 (P = 0.046) with mean increases of 12.7 mg/L, [95CI, 9.7, 15.7] and 12.0 mg/L [95CI, 0.2, 23.8] respectively in low copper medium. Growth in high copper medium suppressed the MET 3/14/16 dependent changes in total SO$_2$ accumulation. In the absence of MET 3/14/16 over-expression, growth in high copper increased total SO$_2$ accumulation in AWRI 4052 only (mean increase = 13.3 mg/L [95CI, 1.9, 25.5], P = 0.024).

The observed MET 3/14/16 dependent increase in total SO$_2$ production in low copper medium indicates that the modifications introduced into these strains successfully increase flux through the sulfate assimilation pathway. However, the data also suggests that copper may suppress either the activity of MET 3/14/16 or efflux of SO$_2$ via *SSU1*.

Total H$_2$S production could only be measured in low copper medium due to complexation between H$_2$S and copper in the high copper condition. In standard defined medium AWRI 4052 produced more H$_2$S than AWRI 3471 (mean diff = 5.7 mg/L, [95CI, 3.8, 7.5], P = 0.001). There was strong evidence (P = 0.0004) that over-expression of MET 3/14/16 increased total H$_2$S production in AWRI 3471 (mean diff = 20.3 mg/L [95CI, 15.2, 25.5]). In AWRI 4052 there was a smaller increase in MET 3/14/16 dependent total H$_2$S production (4.7 mg/L, [95CI, 1.5, 7.7], P = 0.01) (Fig 6D). This result suggests that *SSU1* over-expression constricts flux of sulfur through to H$_2$S.

It should be noted that the parent of AWRI 3471 carries a mutation in *MET2* (R301G) that decreases H$_2$S production, presumably as a result of an improvement in the efficiency of H$_2$S condensation with O-acetyl-homoserine [53]. This mutation is present in both AWRI 3471 and AWRI 4052 and explains the almost complete lack of H$_2$S production in the AWRI 3471 [NatR$^+$] empty vector strain.

Overall, there is no evidence that H$_2$S limitation is a causative factor of copper sensitivity in AWRI 4052.

## Sulfur (SO$_4$) limitation increases copper sensitivity in *SSU1* over-expressing yeast

If *SSU1* over-expression induces sulfur limitation, then the growth of AWRI 4052 should also be compromised in a medium with lower concentrations of SO$_4$. The sensitivity of both AWRI 3471 and AWRI 4052 to low SO$_4$ concentration was evaluated in a defined medium containing a decreasing series of SO$_4$ concentrations (Fig F in S1 Text). This initial screen gave no indication that there were differences in the sensitivity of the two strains to SO$_4$ limitation. However, the trial did suggest a threshold concentration of SO$_4$ (20 mg/L, 208 µmol/L) below which growth was increasingly limited. This SO$_4$ threshold concentration is similar to the concentrations used in previous studies on SO$_4$ limitation [45,54].

Although there was no evidence that *SSU1* over-expression increased sensitivity to SO$_4$ limitation in otherwise replete medium, it is possible that copper could exacerbate the effect of *SSU1* over-expression. This idea was examined by first comparing the effect of increasing copper concentrations on AWRI 3471 and AWRI 4052 growing in defined medium containing a threshold concentration of SO$_4$ (20 mg/L). In these experiments, both growth and fermentation progress were monitored.

AWRI 4052 was highly sensitive to copper in an SO$_4$ limited environment, with as little as 2 mg/L of copper suppressing growth and impeding sugar utilization. A copper concentration of 10 mg/L almost completely abolished sugar utilisation (Fig G in S1 Text). In contrast, there was no evidence for an effect of copper at concentrations up to 10 mg/L on the growth of AWRI 3471 and only a minor perturbation of fermentation progress by 10 mg/L of copper in SO$_4$ limited medium (20 mg/L SO$_4$).

To determine whether AWRI 4052 was more sensitive to copper when experiencing sulfur limitation, we compared its growth and fermentation progress at three copper (0.25, 4.0 and 6.0 mg/L) and two SO$_4$ (20 and 195 mg/L) concentrations (Fig 7).

Analysis by two-way ANOVA found no evidence that the final biomass concentration of AWRI 4052, estimated on day 13, was affected by sulfur limitation alone (P = 0.16) or by interaction with copper (P = 0.12). Copper concentration (P = 0.003) had the largest effect on the

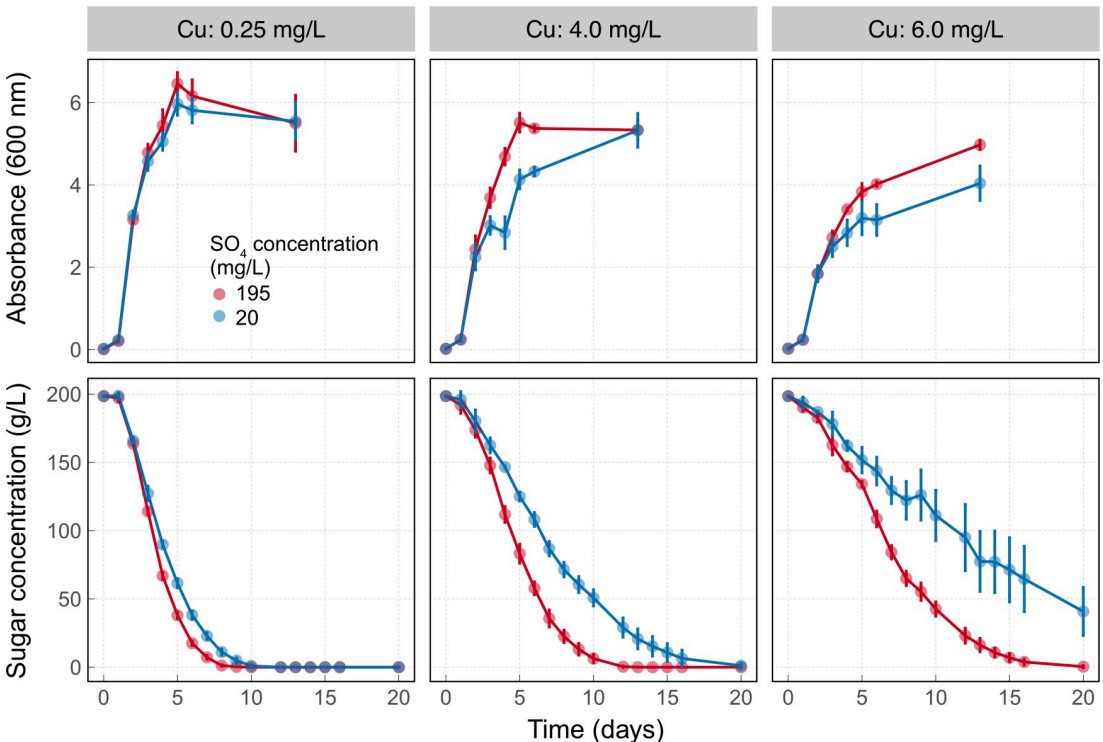

**Fig 7. The combined effect of copper concentration (0.25, 4 and 6 mg/L) and SO$_4$ concentration (20 and 195 mg/L) on the growth and fermentation kinetics of the yeast strain AWRI 4052.** Points show the mean of three replicates with error bars indicating standard deviation.

final biomass concentration accounting for 50% of the total variation in biomass. However, there was evidence that an interaction between copper and SO$_4$ concentration delayed growth. A mean decrease in absorbance of 1.0 [CI 95, 0.3, 1.7] and 0.9 [CI 95, 0.2, 1.6] in SO$_4$ limited medium containing 4 mg/L and 6 mg/L of copper, respectively relative to SO$_4$ replete medium was observed on Day 6.

There was much stronger evidence for an effect of an interaction between copper and SO$_4$ limitation on fermentation time (P < 0.0001), defined here as the time required for the residual sugar concentration to reach 1 g/L. SO$_4$ limitation increased fermentation times by an average of 2 days [95 CI, 0, 4.2] in a low copper medium. The mean difference in fermentation time increased to 5.5 days [95 CI, 3.3, 7.6] in SO$_4$ limited medium containing 4 mg/L copper. Fermentations containing 6 mg/L of copper were not complete on day 20, with 41 g/L (SD 18) of residual sugar at this time. As a result, estimates of the mean difference in fermentation time of 11.5 days [95 CI, 9.1, 13.8] are based on a modelled fermentation. In contrast, the growth of AWRI 3471 in SO$_4$ limited medium was largely unaffected by 10 mg/L copper and sugar utilisation was only slightly delayed (Fig G in S1 Text).

## Conclusions

While copper and SO$_2$ tolerance in *S. cerevisiae* have well described genetic underpinnings, knowledge of *CUP1-2* amplification status was a poor predictor of copper tolerance in wine yeast. Through bulk segregant analysis of strains differentially tolerant to copper this study identified *SSU1* over-expression in SO$_2$ tolerant wine yeast as causative factor in copper sensitivity. The contribution of *SSU1* over-expression to copper sensitivity was validated through

reciprocal hemizygosity analysis. Without an obvious genetic interaction between *SSU1* and *CUP1* or other genes associated with copper tolerance, transcriptomic and proteomic data implicated sulfur limitation in the negative association between the two traits. Over-expression of genes upstream of SO$_2$ in the sulfate assimilation pathway did not improve fermentation performance metrics for an *SSU1* over-expressing strain and did not provide evidence of a role for H$_2$S metabolism in copper sensitivity. That sulfur limitation was involved in copper sensitivity in high *CUP1*-copy number strains was demonstrated experimentally with fermentations using sulfate limited medium.

The findings demonstrate an evolutionary trade-off between SO$_2$ and copper tolerance in yeast and suggests that selection for SO$_2$ tolerance, either directly or inadvertently through the agricultural or oenological application of SO$_2$, could be an additional driving force for continued amplification of the *CUP1-2* locus. It also has important practical implications for strain development, indicating that a less forcefully driven *SSU1* gene may decrease the metabolic burden in commercial yeast strains. Indeed, natural variation in *ECM34(p)* exists and less aggressive versions should perhaps be considered during selection and breeding of commercial yeasts.

## Methods and materials

### Yeast strains and culturing

The strains used in this work are described in Table 1 or in [36] in the case where fitness-based data is presented. Strains used in fitness-based work have been sequenced, and the details of their relationship to other wine yeasts have been described [55]. All strains are available from AWRI Wine Microorganism Culture Collection (AWMCC) and are reported according to their AWMCC identifiers. Strains were maintained on YPD agar (1% w/v yeast extract, 2% w/v peptone and 2% w/v D-Glucose). Experiments were performed in a defined medium [7], the composition of which resembles a Chardonnay juice [7]. Briefly, the defined medium composition consisted of (per litre): glucose 100 g, fructose 100 g, citric acid 0.2 g, malic acid 3 g, tartaric acid 2.5 g, K$_2$HPO$_4$ 1.1 g, MgSO$_4$ .7H$_2$O 1.5 g, CaCl$_2$ .2H$_2$O 0.4 g, H$_3$BO$_3$ 0.04 g, proline 0.84 g, nitrogen as ammonium and amino acids to 300 mg N/L of yeast assimilable nitrogen (YAN), trace elements stock solution 1 mL, vitamins solution 1 mL. Copper was added from a 40 g/L stock solution of CuSO$_4$.5H$_2$O to a 10 mg/L final concentration of copper ion unless otherwise indicated. Low sulfate medium was created by using a combination of 0.05 g/L MgSO$_4$.7H$_2$O and 0.4 g/L MgCl.6H$_2$O as a replacement for 0.2 g/L MgSO$_4$.7H$_2$O.

**Table 1. Yeast strains used in this work.**

| Culture collection identifier | Common name | Description |
|---|---|---|
| AWRI 3019 | 796 | AWRI 796 *ho*::barcode |
| AWRI 3029 | L2056 | AWRI 1487 *ho*::barcode |
| AWRI 3032 | Vin13 | AWRI 1537 *ho*::barcode |
| AWRI 3471 | | AWRI 3019 haploid *MATα* |
| AWRI 3807 | | AWRI 3029 haploid *MATa* |
| AWRI 3470 | | AWRI 3032 haploid *MATa* |
| AWRI 3811 | | AWRI 3471 x AWRI 3807 |
| AWRI 3001 | | AWRI 3471 x AWRI 3470 |
| AWRI 3901 | | AWRI 3811 *ssu1Δ*::*KanMX4* chrXVI |
| AWRI 3902 | | AWRI 3811 *ssu1Δ*::*KanMX4* chrVII-XVI |
| AWRI 4052 | | AWRI 3471 *ssu1(pr)Δ*::ECM34(pr) |

Fermentations were conducted in 100 mL vessels as described in [36] or in microtiter plates as described in [56]. Fermentation vessels were stirred at 250 rpm and incubated at 17˚C. All treatments were performed in triplicate except for the screening of spores for copper tolerance, for which n = 2.

## Molecular characterisation of spores

All primers used in this work are listed in Table A in S1 Text. Spores of strains AWRI 3019, AWRI 3029 and AWRI 3032 were generated by inducing sporulation on 1% w/v potassium acetate agar plates. A sample of the starved cultures was treated with zymolyase, and spores were dissected using a micromanipulator (Singer Instruments) and asci arrayed onto YPD agar. All viable spores were screened for the presence of a barcode using Illumina sequencing primers (Illum_read1 and Illum_read2) with the following amplification conditions; 95˚C for 1 min and 35 cycles of 95˚C for 10 s, 60˚C for 5 s, 72˚C for 20 s. Spores were also assessed for their mating type status according to the method of Illuxley et al [57] using primers Primer_-MAT, Primer_MATa and Primer_MATalpha under the following conditions; 92˚C for 1 min and 30 cycles of 92˚C for 10 s, 58˚C for 30 s, 72˚C for 20 s. Spores that contained a barcode and were either *MATα* (if derived from AWRI 3019) or *MATa* (if derived from AWRI 3029 or AWRI 3032) were assessed for their ability to grow in a defined medium containing 10 mg/L copper. One spore of AWRI 3019 was isolated that contained a molecular barcode (MBC), was *MATα* and was copper tolerant; this isolate was designated AWRI 3471 (*MATα*, *ho*::barcode). One spore from each of AWRI 3029 (*MATa*, *ho*::barcode) and AWRI 3032 (*MATa*, *ho*::barcode) was isolated that contained an MBC, was *MATa* and was copper sensitive; these spores were designated AWRI 3807 and AWRI 3470, respectively.

## Mating of yeast

Copper tolerant (AWRI 3471) and sensitive (AWRI 3807 and AWRI 3470) spores were mated to produce diploid yeast that contained barcodes from each of the respective parents. These yeasts, their culture collection identifiers and descriptions are listed in [36]. Mating was undertaken by growing each parent in YPD overnight at 28˚C, adjusting the culture densities to equivalency (absorbance at 600 nm of 1.5) and mixing equal volumes of each culture. Microscopic examination of mixed cultures was undertaken until zygotes were observed (approximately 2 hours). Potential zygotes were isolated using a micromanipulator (Singer instruments). The isolates were screened for both *MATa* and *MATα* mating-type genes by PCR using the conditions described above. Positive colonies were again subcultured, clonal individuals selected, and the presence of both mating-type genes confirmed. Two diploids were isolated in this way. AWRI 3811 was the product of a cross between AWRI 3471 and AWRI 3807. AWRI 3001 was the product of a cross between AWRI 3471 and AWRI 3470.

The two diploid strains were sporulated, and spore dissections were performed to isolate 74 spores and 118 spores from AWRI 3001 and AWRI 3811, respectively. The copper sensitivity of the spores was characterized in 200 μL microplate cultures (AWRI 3001 spores) or 100 mL flask cultures (AWRI 3811 spores) in a defined medium containing copper concentrations of either 0.25 mg/L (control) or 10 mg/L (high copper treatment). Duplicate ferments were used for the determination of spore sensitivity to copper. Growth was assessed by measuring absorbance at 600 nm after 52 h (AWRI 3001 spores) or 72 h (AWRI 3811 spores). Spores were defined as copper sensitive if the mean difference in absorbance (600 nm) between growth in low and high copper medium after 72 h was greater than 2.5 and $P < 0.05$ in an unpaired T-test. Fig A in S1 Text shows the mean difference in absorbance of spores and indicates the

pools to which they were assigned. Spore phenotype raw data and associated analysis is given in [41, file T03].

## Preparation of pooled DNA for QTL analysis

Overnight YPD cultures of individual spores were used as a source of genomic DNA. Genomic DNA was isolated from all spores derived from AWRI 3001 and AWRI 3811 using a Gentra Puregene Yeast/Bact kit (Qiagen) according to the manufacturer's instructions, except that 6 μL of lytic enzyme was used. DNA concentrations were determined using a Qubit fluorometer (Thermo Fischer Scientific). Two pools of DNA were prepared from each set of spores derived from AWRI 3001 and AWRI 3811. One pool (pool 1) was comprised of DNA from spores, for which there was no evidence of a difference in the absorbance attained when spores were grown in low or high copper media (P > 0.05). A second pool (pool 2) was comprised of DNA from spores for which the difference in absorbance between low copper and high copper grown cells was greater than 2.0 absorbance units and if an unpaired T-test produced a P value of < 0.05. In each case, 300 ng of DNA from each spore was added to the pool such that all spores were equally represented within the pool.

## *SSU1* promoter replacement with the *ECM34* promoter to create AWRI 4052 *ssu1*(pr)Δ::p*ECM34(pr)*

*SSU1* promoter replacement in the AWRI 3471 strain was undertaken using the approach of Storici and Resnick [58] using *natMX* selective and *GIN11* counter selective markers as previously described [59]. An amplicon was produced from a pAG25-GIN11 plasmid (CORE1 cassette) using CORE1_Ampl-F/ CORE1_Ampl-R primers carrying 50 bp sequences homologous to the *SSU1* genomic target site using the following PCR conditions 95˚C for 1 min and 30 cycles of 95˚C for 10 s, 58˚C for 15 s, 72˚C for 3 min. The obtained PCR product was transformed into AWRI 3471 using an adaptation of the LiAc method described by Gietz et al [60]. Cells were recovered for 2 hours in liquid YPD medium at 30˚C, and transformants isolated on YPD agar plates containing 100 μg/mL clonNAT. Successful integration of the CORE cassette was confirmed using PCR with Chr16_SSU1_Prom-F/ Chr16_SSU1_Prom-R primers under the following conditions 95˚C for 1 min and 30 cycles of 95˚C for 10 s, 58˚C for 15 s, 72˚C for 3 min. A single colony isolate with a confirmed CORE1 integration was then transformed, as described above, with an amplicon of 1349 bp containing the 1005 bp promoter region of *ECM34* from yeast strain AWRI 3807 flanked by 156 bp and 188 bp sequences homologous to up- and down-stream regions of the replaced *SSU1* promoter. After YPD recovery, transformations were washed twice in sterile deionized water and plated onto YNB agar plates containing galactose as a sole carbon source to enable *GIN11* expression (counter selection). CORE cassette replacement in isolates from galactose plates was confirmed by PCR using primers Chr16_SSU1_Prom-F/ Chr16_SSU1_Prom-R and the following conditions: 95˚C for 1 min and 30 cycles of 95˚C for 10 s, 58˚C for 15 s, 72˚C for 3 min.

## Over-expression of *MET3*, *MET14* and *MET16* in AWRI 3471 and AWRI 4052

The genes in the upper branch of the sulfur assimilation pathway were over-expressed via replacement of their native promoters with constitutively expressed yeast promoters [61]. 550 bp regions of chosen promoters; *FBA1*, *PGK1* and *PGI1*, were used to over-express *MET3*, *MET14* and *MET16* genes respectively. All genes were designed to sustain 200 bp of their native terminators. The element containing over-expressed genes was designed using

SnapGene software, and subsequently divided into 3 fragments, 1655 bp, 2289 bp and 1836 bp in length which were synthesized by Decode Science. Each fragment carried 100 bp homologous overlapping sequence which allowed the assembly of the fragments and their introduction into the centromeric vector p416-natR using yeast homologous recombination machinery. To achieve this the strains AWRI 3471 and AWRI 4052 were transformed with empty p416-natR vector (control strains), and 3 fragments plus linearized p416-natR vector DNA (1 μg each) using LiAc / heat shock transformation protocol. Transformations were recovered for 2 hours in liquid YPD medium at 30˚C and plated onto YPD agar plates containing 100 μg/mL clonNAT. Strains derived from single colonies growing on selection plates were confirmed using PCR with p416hy-FR1-F/ FR2-URA3-p416-R primers under following conditions: 95˚C for 1 min and 30 cycles of 95˚C for 10 s, 58˚C for 15 s, 72˚C for 4 min.

## Sample preparation and genomic sequencing for bulk segregant analysis

DNA from individual yeast or pooled genomic extracts were brought to a concentration of 5 ng/μL. Sequencing libraries were prepared using an Illumina Nextera XT kit (AWRI 3001 and derived spores) or a TrueSeq Nano kit (AWRI 3811 and derived spores). Sequencing was performed on an Illumina MiSeq v3 using a 2 x 300 bp dual indexing kit by the Ramaciotti Centre for Genomics (University of New South Wales, Sydney, Australia). Raw data was quality trimmed and mapped as described previously [55].

## Genomic copy number estimation by quantitative PCR (qPCR)

Genomic DNA was extracted from each of the barcoded strains as described above, their concentration equalised to 10 ng/μL and then further diluted to 0.25 ng/μL. qPCR experiments were performed on a CFX Real-Time PCR detection system (BioRad, Hercules, California) using KAPA SYBR FAST qPCR Kit (Roche Life Sciences, North Ryde, NSW) and 200 nM specific primer pairs; Barcode_F and Barcode_R to amplify the barcode inserted into the HO locus (described above) and CUP1F_SS and CUP1R_SS for amplification of CUP1 (see Table A in S1 Text). The primers were designed to give similar product sizes, and the PCR efficiency of each primer pair (Eff) was evaluated by the dilution series method using genomic DNA extracted from strains AWRI 3032 and AWRI 3019 [36]. *CUP1* gene copy number was estimated using the R package qpcR [62] relative to the barcode copy number (1 copy per cell). The total number of *CUP1* copies in each strain was estimated over a series of 8 experiments in which a dilution series of AWRI 3019 DNA (0.004, 0.04, 0.4, 4.0, 40.0, 60.0 ng) was amplified with Barcode_F and R primers to generate a standard curve. The single gene copy number (CN) per ng of diploid yeast genome was estimated using the formula (equation 1) provided in Brankatschk et al [63]. The linear regression of log concentration (DNA concentration of the standard) and amplification Ct values was calculated and used to estimate sample *CUP1* copy number using the "calib" function, drawing on second derivative threshold cycles calculated by modlist and derived using an L4 model. Quantitative PCR was performed using 4 μL of template DNA (0.25 ng/μL) under the following conditions; 95˚C for 1 min, then 40 cycles of 95˚C for 10 s, 60˚C for 5 s, 72˚C for 20 s.

## Gene expression analysis

**Conduct of fermentation.** The yeast strains AWRI 3471 and AWRI 4052 were initially grown in YPD overnight at 28˚C. Overnight cultures were used to inoculate a half-strength defined medium (diluted with water) which was grown for a further 24 h at 28˚C. 500 μl of overnight half-strength defined medium culture was used to inoculate 100 mL of standard defined medium or standard defined medium containing 10 mg/L (0.157 mM) copper.

Samples were taken for RNA extraction 48 h post-inoculation. The absorbance values and estimated cell concentrations of the cultures at the time that the samples were taken are reported in [41, file T06].

**RNA extraction and sequencing.** Cell pellets were prepared from samples by centrifugation for 1 min at 16,000 × g and the supernatant removed. Tubes containing cell pellets were placed in dry ice for 5 min before being stored at -80°C. Frozen cells were thawed rapidly in 600 μL of lysis buffer (PureLink RNA mini kit (Invitrogen) and disrupted using 200 mg of acid-washed glass beads (Sigma G8772) with processing in a Bertin Precellys Evolution for 3 x 20 s at 6800 rpm with 30 s pause on ice between cycles. RNA was extracted from the homogenate using a PureLink RNA mini kit according to the manufacturer's instructions. RNA concentration was estimated using a Qubit fluorometer (Thermo Fischer) and quality assessed using a Tapestation (Agilent), providing RINe values between 9.0 and 9.6. Total RNA was prepared for sequencing using a stranded mRNA-seq prep kit (Illumina) and sequenced using a 1 x 75 bp on a NextSeq 500 using a high-output flow cell at the Ramaciotti Centre for Genomics (Sydney, Australia).

**Data processing and differential gene expression analysis.** Illumina single-end reads were quality trimmed using Trimmomatic v.0.38 [64]. The creation of a genome index and mapping of the Illumina reads to the reference genome of *S. cerevisiae* s288c was performed using STAR v.2.7.3a [65]. Prior to mapping, the paralog of *CUP1-1*, *CUP1-2*, was masked to avoid multi mapping reads. Counting of reads mapping to each genomic feature was performed using featureCounts v.2.0.0 [66]. Read count tables were imported into R [67], features with 0 counts in all samples were removed, and differential gene expression analyses were performed using the DESeq2 package v.1.24.0 [68] with default parameters (sample-wise size factor normalization, Cox-Reid dispersion estimate and the Wald test for differential expression), comparing each strain (AWRI 3471 and AWRI 4052) and treatment (0.25 mg/L and 10 mg/L copper) against the corresponding control. Features with a Log$_2$ fold-change (Log$_2$FC) of $1 < \text{Log}_2\text{FC} < -1$ and an adjusted p-value $< 0.005$ were considered for further analysis. Over-Representation Analysis (ORA) of differentially expressed gene sets was undertaken using Metascape 3.5 vDec 18, 2021 [69]. As an independent variable in the experiment *SSU1* was omitted from the gene list submitted to the Metascape ORA. Mapping of differentially expressed gene sets to transcription factors was done using Yeastract+ [70] only retrieving transcription factors for which there was DNA binding and expression evidence.

## Label-free proteomic analysis of yeast strains

**Sample extraction, protein reduction and alkylation.** Yeast cells were harvested and stored as described above. Cell pellets were thawed gently, and 100 μL of each was washed with 1 mL 1 x TBS buffer, vortex mixed briefly, centrifuged at 5,000 × g for 2 min and the supernatant removed. This was repeated before a 50 μL aliquot was placed in a fresh tube with 200 μL of RIPA buffer (2 x concentration with protease inhibitor cocktail and dithiothreitol (DTT, 20 mM) added). Samples were vortexed briefly and heated at 95°C for 5 min. Following this, they were cooled and placed in the Diogenode Bioruptor on high for 10 min. The samples were then further reduced at 56°C for 15 min, followed by alkylation with iodoacetamide (IA, 55 mM) in the dark at room temp for 30 min. Fresh DTT was then added to the reaction to quench the IAM. The samples were centrifuged at 5,000 × g for 1 min, and the supernatant (114 μL) was removed to a fresh tube along with 1 μL of the pelleted cells and debris.

**Protein quantitation and tryptic digestion.** Six volumes (690 μL) of cold acetone were added to each 115 μL sample, mixed briefly and placed at -20°C overnight to precipitate. The samples were centrifuged at 20,500 × g at -9°C for 20 min, and the supernatant was removed.

The pellets were washed with 0.5 mL of 80% cold acetone, centrifuged again at 20,500 × g for 10 min and the pellets air-dried for 5 min. They were then resuspended in 10 μL of 6 mol/L guanidine/25 mmol/L ammonium bicarbonate solution with vortex mixing, sonication and trituration with a micropipette, followed by the addition of 90 μL of 25 mmol/L ammonium bicarbonate and further mixing and sonication. Protein concentration in each sample was then determined using an EZQ Protein Quantitation Kit (Thermo Fisher Scientific, USA), according to the manufacturer's protocol.

For each sample, 40 μg total protein was digested at 37˚C overnight in 25 mmol/L ammonium bicarbonate/3% acetonitrile using 1 μg of sequencing grade porcine trypsin (Promega, USA) in a total volume of 150 μL. The enzymatic digestion was stopped with the addition of trifluoracetic acid (TFA) to 0.5%. The tryptic peptides were cleaned up using Pierce C18 Spin Columns (Thermo Fisher, USA), following the manufacturer's protocol. Eluted peptides were dried in a Speedvac to about 2 μL, and then all samples were made up to 25 μL with the addition of 3% acetonitrile. The relative concentration of the samples was determined using a Nanodrop spectrophotometer at 205 nm, followed by the addition of 0.5 μL of 10% TFA to each 25 μL sample to give a final concentration of ~3% acetonitrile/0.2% TFA.

**Data acquisition.** Peptides were separated using an ultiMateTM3000 RSLC nano liquid chromatography system (Thermo Fischer Scientific, USA) coupled online to a timsTOF Pro mass spectrometer (Bruker Daltonics, Germany) for analysis using the default parameters in Data-Independent Acquisition–Parallel Accumulation Serial Fragmentation (DIA-PASEF) long-gradient mode. Reverse-phase chromatography was performed using a 25 cm, 75 μm ID Aurora C18 nano column with an integrated emitter (Ion Opticks, Australia). The peptides (~200 ng) were eluted using a 125 min gradient from 0% to 37% buffer B (0.1% formic acid in acetonitrile) at a rate of 400 nL min$^{-1}$. Buffer A consisted of 0.1% aqueous formic acid.

**Data processing and analysis.** Raw data files from each sample generated on the timsTOF Pro mass spectrometer were processed using the software package MaxQuant1 v2.0.3.0 (Max Planck Institute of Biochemistry, Germany). The data were searched against the MaxQuant discovery library and FASTA database for *Saccharomyces cerevisiae* with the following parameters: variable modifications–deamidation (N/Q), oxidation (M); fixed modification–Carbamidomethyl (C); enzyme–trypsin; missed cleavages– 3, using standard Tims MaxDIA parameters. The first and main search tolerances were set to 40 ppm and 20 ppm, respectively. Proteins with a false discovery rate (FDR) of ≤ 1% were reported. MaxQuant LFQ values were imported into R and analysed using the DEP package v. 1.19.0 [71]. The data was initially filtered for proteins identified in all replicates of at least one treatment and normalised by variance stabilizing transformation [72]. Data imputation for missing values was performed using random draws from a Gaussian distribution centred around a minimal value [73]. Differential enrichment analysis was performed using limma v. 3.48.3 [74] and proteins with a $1 <$ Log$_2$FC $< -1$ and an adjusted p-value $< 0.005$ were classified as differentially enriched.

## Detection of translocations related to SO$_2$ tolerance

Genomic DNA was extracted from each of the barcoded strains as described above, and their concentration equalised. Determination of Chr VIII::XVI and ChrXV::XVI translocations status was performed by presence/absence of PCR products following amplification of genomic DNA using primers listed in Table A in S1 Text. The primers are based on those designed by Pérez-Ortín et al [27] and Zimmer et al [28]. Table B in S1 Text lists the primer pairs used to detect specific chromosomal rearrangements and the fragment size used to score the translocation. PCR was performed using 2 μL of template DNA (5 ng/μL) under the following conditions; 95˚C for 2 min, then 35 cycles of 95˚C for 10 s, 55˚C for 40 s, 72˚C for 20 s.

## Analysis of juice and medium composition

The determination of free and total SO$_2$ was undertaken by the Australian Wine Research Institute Commercial Services laboratory using a discrete analyser (Thermo Gallery). Reagents and absorbance wavelengths for the determination of free and total SO$_2$ in this method were pararosaniline and formaldehyde (575 nm), and 5,5′-dithio-bis-2-nitrobenzoic acid (412 nm), respectively. Glucose and fructose concentrations were determined enzymatically [75] with adaptations as described by Vermeir et al. [76] for the performance of 200 μL assays in 96-well microtiter plates. Other compositional analyses were undertaken by The Australian Wine Research Institute (AWRI) Analytical Service [International Organization for Standardization 17025 accredited laboratory, Adelaide, SA, Australia]. Metal ion concentrations were determined as described by Wheal et al [77] on a Perkin-Elmer (Waltham, USA) inductively coupled mass spectrometer model Nexion 350D with the following settings: RF power 1400W, plasma argon flow rate 18 L/min, nebuliser flow rate 0.75–0.80 L/min. Yeast assimilable nitrogen (YAN) was determined using a combination of the NOPA assay for the determination of free amino nitrogen [78] and enzymatic determination of ammonium. YAN was calculated as follows: YAN (mg/L) = ammonium × 0.825 (mg/L) + FAN (mg/L).

## Statistical analyses

Data relating to analyte concentration or ferment duration were subjected to one-way ANOVA using the aov function in R (version 4.2.1) to determine whether means differed with regard to treatment (n = 3 for all treatments). If ANOVA P values were less than 0.05 a multiple comparison with respect to treatment was undertaken using the function HSD.test (agricolae) to determine the mean difference, upper and lower confidence intervals for the contrasts at alpha = 0.05.

## Supporting information

**S1 Text. Fig A: The mean difference in growth of spores in defined medium containing 0.25 mg/L relative to medium containing 10 mg/L of copper.** Spores were isolated from yeast strains AWRI 3001 and AWRI 3811. Growth of spores was assessed in microplates and 100 mL cultures for AWRI 3001 and AWRI 3807 respectively. Yeast growth was assessed as absorbance at 600 nm after 72 h incubation at 17˚C. Bars show the mean difference in absorbance (600 nm) with error bars indicating the 95% confidence interval (n = 2). Bars are coloured by the pool to which each spore was assigned for bulk segregant analysis. **Fig B: Comparison of haploid *CUP1* copy number variation (CNV) as estimated by Steenwyk et al [15] and absolute CNV estimated by Onetto (this work) in yeast strains common to both studies.** CNV estimated by Onetto et al are the mean of at least three independent estimates with error bars showing standard deviation. **Fig C: Growth of spores isolated from yeast strains A) 3019, B) 3029 and C) 3032 in defined medium containing 0.25 mg/L (blue) or 10 mg/L (red) of copper.** Yeast growth was assessed as absorbance at 600 nm after 48 h (A and C) or 72 h (B) incubation at 17˚C. Error bars show the mean of 3 (B) or 4 (A and C) replicates. Growth of the diploid parent for each set in both conditions is also shown (parent). **Fig D: Over-representation analysis of transcripts with differential abundance in an *SSU1* over-expressing strain.** Transcript abundance in AWRI 4052 was compared to transcript abundance in the cognate unmodified strain AWRI 3471 growing in defined medium containing 10 mg/L copper. Transcript classes that were over-represented in the *SSU1* over-expressing strain are shown. **Fig E: Over-representation analysis of proteins with differential abundance in an *SSU1* over-expressing strain.** Protein abundance in AWRI 4052 was compared to protein abundance in the cognate unmodified strain AWRI 3471 growing in defined medium

containing 10 mg/L copper. Protein classes that were over-represented in the *SSU1* over-expressing strain are shown. **Fig F: Effect of sulfate concentration on the growth of yeast strains AWRI 3471 (red) and AWRI 4052 (blue).** Growth was determined by measuring absorbance at 600 nm. Yeast strains were grown in defined medium using 100 mL fermentation vessels. The height of each bar is the absorbance reading for a sample (n = 1). Data collected on day 4 and day 5 are shown. This information was used to estimate a suitable concentration of sulfate to use in subsequent experimental work. **Fig G: The effect of increasing copper concentrations on the growth and fermentation kinetics of yeast strains AWRI 3471 and AWRI 4052.** Yeast were grown in defined medium containing an estimated 20 mg/L $SO_4$. Growth was followed by measuring absorbance at 600 nm. Sugar concentration is the sum of glucose and fructose concentrations measured enzymatically. Points show means (n = 3) and error bars show standard deviation. **Table A: Primers used in this work.** Primers used for chromosomal translocation status are taken from Zimmer et al [28]. **Table B: Primer pairs used to detect chromosomal rearrangements. Table C: Effect of Copper concentration and plasmid containing *MET 3/13/16* genes on $SO_2$ production by AWRI 3471.** $SO_2$ concentration is given as the mean of three replicates (Mean $SO_2$) with standard deviation (sd) shown. Two-way ANOVA analysis of was conducted with yeast strain, copper concentration and yeast * copper investigated as factors at alpha = 0.05. The table shown gives the results of the ANOVA and a TUKEY multiple pairwise comparison evaluating the magnitude of differences between pairs of treatments. **Table D: Effect of Copper concentration and plasmid containing *MET 3/13/16* genes on $SO_2$ production by AWRI 4052.** $SO_2$ concentration is given as the mean if three replicates (Mean $SO_2$) with standard deviation (sd) shown. Two-way ANOVA analysis was conducted with yeast strain, copper concentration and yeast * copper investigated as factors at alpha = 0.05. The table shows the results of the ANOVA and TUKEY multiple pairwise comparison evaluating the magnitude of differences between pairs of treatments.
(DOCX)

## Acknowledgments

For genome sequencing the authors would like to thank the Ramaciotti Center for Genomics which is funded through Bioplatforms Australia Pty Ltd (BPA), a National Collaborative Research Infrastructure Strategy (NCRIS). Proteomic data acquisition was obtained with support of the Adelaide Proteomics Centre at The University of Adelaide, in partnership with the South Australian Health and Medical Research Institute Proteomics Core Facility. Special thanks to Tara Pukala for facilitating that analysis. Thanks to Markus Herderich for critical review of the manuscript.

## Author Contributions

**Conceptualization:** Anthony R. Borneman, Simon A. Schmidt.

**Data curation:** Simon A. Schmidt.

**Formal analysis:** Cristobal A. Onetto, Anthony R. Borneman, Simon A. Schmidt.

**Funding acquisition:** Simon A. Schmidt.

**Investigation:** Cristobal A. Onetto, Dariusz R. Kutyna, Radka Kolouchova, Jane McCarthy.

**Methodology:** Cristobal A. Onetto, Dariusz R. Kutyna, Radka Kolouchova, Jane McCarthy, Anthony R. Borneman, Simon A. Schmidt.

**Project administration:** Simon A. Schmidt.

**Software:** Cristobal A. Onetto, Simon A. Schmidt.

**Supervision:** Simon A. Schmidt.

**Visualization:** Simon A. Schmidt.

**Writing – original draft:** Simon A. Schmidt.

**Writing – review & editing:** Cristobal A. Onetto, Dariusz R. Kutyna, Jane McCarthy, Anthony R. Borneman, Simon A. Schmidt.

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
