## [Decision Letter · Decision Letter 0]

4 Jan 2023

Dear Dr Schmidt,

Thank you very much for submitting your Research Article entitled 'SO2 and copper tolerance exhibit an evolutionary trade-off in Saccharomyces cerevisiae.' to PLOS Genetics.

The manuscript was fully evaluated at the editorial level and by independent peer reviewers. The reviewers appreciated the attention to an important topic but identified some concerns that we ask you address in a revised manuscript.

We therefore ask you to modify the manuscript according to the review recommendations. Your revisions should address the specific points made by each reviewer.

Yours sincerely,

Justin C. Fay

Academic Editor

PLOS Genetics

Geraldine Butler

Section Editor

PLOS Genetics

All three reviewers were positive but brought up important issues that would improve the manuscript. I would like to emphasize one point. The relationship between CUP1 copy number and resistance should be clarified in relation to prior work. Is this due to the copper concentration, the medium to which copper was added or some other factor. The significance of your findings is related to a careful explanation of how copper resistance was measured. Rather than citing reference 7, please include the medium and copper concentration, and the formulation (since g/mL differs depending on the form of copper used). A reasonable but unanswered question that a reader might have is whether SSU1 genotype affects copper resistance in rich or complete medium - probably the most commonly used assays.

Reviewer's Responses to Questions

**Comments to the Authors:**

Reviewer #1: This manuscript demonstrates correctly the genetic interaction between SSU1 translocation and CUP1 copy number on copper tolerance. The molecular mechanisms behind this interaction are still to be elucidated. However, this manuscript rigourously reject many potential causes and shows a link with sulfur limitation.

This work is original and of importance for researchers in the field.

Before recommending publication I would like to have precisions regarding the following comments :

General major comments :

I am confused with these sentences :

L99 : "The diverse CUP1 copy number variation among strains in the wine yeast clade is consistent with the previously observed fitness variation in high copper medium [34]. However, there was a poor correlation between CUP1 copy number and relative strain fitness in a medium containing 10 mg/L copper (Fig 1)"

I'm not sure to what "consistent" is reffering to, and which concentration of copper medium is considered as high. In ref [34], 10 mg/L copper seems to be also used. Therefore, I don't understand the contrast between the two sentences. I suggest to show the relationship between CUP1 copy number and strain fitness for the two level of copper. This will highlight at wich concentration the trade-off is present.

Regarding the title of the paragraph

(L 92-93)High CUP1 copy number is necessary but not sufficient in determining copper tolerance in wine yeast

Fig 1 is not showing that as some strains appear to be resistant (266 or 1722) without having high CUP1 copy number. As mentioned in the subsequent text, in this condition no link between CUP1 and strain fitnees is found.

The fact that no relation between CUP1 and strain fitness is found should be more discussed as many papers even cited in this manuscript demonstrated a link between CUP1 copy number and copper tolerance.

It may be specific to the condition tested (copper concentration) or to the genetic background (wine yeasts that are enriched in SSU1 translocation).

108 -109 : Poor predictor in this specific condition. As mentioned in the introduction the association is described in other papers.

I have another major concern regarding the GO category as shown in Fig4 panel C.

It seems that many genes are not with correct GO category. For exemple *AUS1* is a sterol transporter and has not the sulfur compound transport GO:0072348 term. I suggest to check this issue and rerun the analysis if necessary.

Minor specific comments

The introduction presents SO2 and copper resistance mechanisms. However, the potential interaction between them is presented in only two sentences L76 to L78. As it is the main subject of the manuscript I suggest to add more detail regarding the data supporting their interaction (ref 33,34) and hypothesis behind that.

S3 fig. L118 in the manuscript is mentioning AWRI796, AWRI 1537 and AWRI 1487, figure shows 3019, 3029 and 3032

169 : It would be easier to get the construction by mentioning directly that AWRI 4052 was generated from AWRI 3471. You can also add panel A in figure quote.

L187 : Fig4 shows that a considerable number of genes associated to sulfate / sulfur GO term have also decreased expression in AWRI 4052. The manuscript only highlight increased expression cases. Comments on other genes would be interesting.

Fig4 panel C. I suggest to mention SSU1 as a control. SSU1 is not part of any of these GO categories ?

Fig4 panel D. Caption is mentioning red and blue colors but it’s orange and blue. You may keep the same color code blue/red used in previous panels when comparing AWRI4052 - AWRI3471 in i) and ii) subpanels and choose a different in iii) iv).

L246 and Fig4 panel D, subpanel ii). Figure would be easier to read with SSU1 labelled on the figure. According to the text in the manuscript it still have increased abundance.

L413 Error!

SO2 concentration is indicated either in mg/L or in µM according to text or figure. Homogenization should be good

L320 S6 fig : I guess that x axis scale is wrong as the manuscript is describing a concentration of 20 mg/L (L320)

L347 % of variance explained would be interesting

Fig 7 is lacking the strain AWRI 3471 as control.

Reviewer #2: This manuscript is reporting a detailed analysis of the tradeoff between copper and SO2 tolerance in S. cerevisiae. The paper is significantly progressing our understanding of the molecular determinants responsible for the mutually exclusive tolerance to these 2 compounds. The work is rigorously done and very convincing. I have only minor suggestions to improve the quality of the paper.

Minor modifications:

INTRODUCTION:

- The introduction is pleasant to read and nicely introduces the general question on the genetic determinants controlling copper and SO2 tolerance in yeast. However, the authors should mention that in addition to translocations, an inversion on chrXVI was reported to also promote sulfite tolerance (PMID: 30859719). Furthermore, it would be worth mentioning that the COM2 regulon has also been involved in the tolerance to sulfur dioxide (PMID: 31799324).

RESULTS:

- In the first paragraph the authors indicate that their qPCR estimates are “consistent” with the copy number estimated from whole-genome sequence data. However, there is ~2 fold difference between the 2 studies across the entire range of values that should at least be commented. In addition, the CUP1 copy number per strain should be carefully verified as there seems to be some inconsistencies between the y-axis of Fig1 and the y-axis of Supp Fig2, which are supposed to represent the same data. For instance, the strain 2914 has nearly 40 copies based on Schmidt Supp Fig 2 while it goes down to only about 32 on Fig 1. Finally, the presence of the barcode that is used to estimate the CUP1 copy numbers in this study should be better recalled than ‘described in [34]’ and it would be also nice to explain how the relative strain fitness was calculated.

- In the second paragraph, we expect to see the growth of spores isolated from the AWRI 796, AWRI 1537 and AWRI 1487 strains but the numbers are different in Supp Fig 3 which makes the story very difficult to follow (3019, 3029 and 3032). The reader has to refer to Table 1 to follow the strain names. I t would be much better to keep a single name all across the main text and in all figures.

- In the third paragraph, for the QTL analysis of the AWRI3471, the authors report a minor QTL at ~1,000,000 bp that is very difficult to locate on Figure 2 because the genome position scale is not readable. In addition, there are many other deviations from the 0.5 expected proportion, in several chromosomes (including the entire chromosome I), that are disregarded without being at all discussed.

- In the last paragraph of the Results (Sulfur limitation increases copper sensitivity in SSU1 over expressing yeast), there might be an issue with the units as the main text is reporting a threshold concentration of SO4 of 20 mg/L while in the Sup Fig 6, this threshold appears to be at 0.02 mg/L. The SO4 concentration in the figure should possibly be expressed in g/L.

- The authors should consider the possibility of including the Supp Fig 7 as a panel of the main Fig 7.

-

METHODS:

- There are 2 Error messages “Reference source not found” that need to be fixed.

Reviewer #3: In this manuscript, Onetto et al. investigate the negative correlation between Cu and SO2 tolerance in S. cerevisiæ. Through segregant QTL-analysis, they identified the genetic variance causing Cu-sensitivity to be at SSU1, which is previously linked with SO2 tolerance. They show that overexpression of SSU1 effect on Cu tolerance is not caused by a weakening of the molecular defence against Cu toxicity. Instead, they show that Cu-sensitivity phenotype is metabolically linked to SO2 limitation. This is an interesting study that shed new lights on the mutual exclusivity of the two important traits that are Cu and SO2 tolerance in yeast. The works are very well conducted, the analyses are exhaustive, and they made a thorough investigation of the SSU1 dependant Cu-sensitivity. Here are minor points I believe would strengthen the paper.

1. I would recommend the authors to explicitly mention the form of copper used in the experiments so reader can fully appreciate the nature of the cell stress.

2. Figures would benefit, in terms of readability, from more detailed annotations of the genotypes. Especially in Fig4 A and B, a small space could be allowed to specify the mutation of SSU1 promotor.

3. In results section, line 173. The authors make the first mention about monitoring fermentation progress. I feel like the rational and explanations for this experiment should be stated more clearly (and the comment also apply to later in the results, line 328, at other mention of fermentation monitoring). Moreover, the implications of this results (delay in fermentation) for the following experiments is lacking.

4. Line 318, the authors refer to S6 fig as an initial screen that suggest a threshold concentration of 20mg/L SO4. However, the S6 fig shows almost unaltered growth for SO4 concentrations ranging 0.03 to 0.2mg/L. Maybe is it due to a mistake in S6 Fig labelling? Or other results are needing to fully support the threshold value?

5. One topic I feel is not developed enough in the manuscript is the SO4 tolerance. Although the demonstration of Cu-sensitivity caused by SSU1 overexpression is strong, it let me wonder whether the system used (swap of endogenous SSU1 promoter for ECM34 promotor) is able to increase SO4 tolerance. I am not sure if such an assay is feasible. But testing SO4 tolerance in AWRI 4052 in comparison to AWRI 3471 would be a great characterization of the strain to better link Cu- and SO4-tolerance traits. If this was already tested in previous work, explicitly mention it would be recommended.

**Have all data underlying the figures and results presented in the manuscript been provided?**

Reviewer #1: Yes

Reviewer #2: Yes

Reviewer #3: Yes

PLOS authors have the option to publish the peer review history of their article (what does this mean?). If published, this will include your full peer review and any attached files.

Reviewer #1: No

Reviewer #2: **Yes: **Gilles Fischer

Reviewer #3: **Yes: **Samuel Plante

---

## [Decision Letter · Decision Letter 1]

6 Mar 2023

Dear Dr Schmidt,

We are pleased to inform you that your manuscript entitled "SO_2_ and copper tolerance exhibit an evolutionary trade-off in *Saccharomyces cerevisiae*." has been editorially accepted for publication in PLOS Genetics. Congratulations!

Yours sincerely,

Justin C. Fay

Academic Editor

PLOS Genetics

Geraldine Butler

Section Editor

PLOS Genetics

Comments from the reviewers (if applicable):

Reviewer's Responses to Questions

**Comments to the Authors:**

Reviewer #1: The manuscript was improved to address the points raised by reviewers. I'm favorable for publication.

Answer to the authors:

Line 117 : “and the a high” a typo was added during rewriting process

S4 Table : I was not able to locate a potential newly updated supplementary file, but in the previous submission S4 Table title mentioned AWRI3471 instead of AWRI4052

Author response: We are not entirely sure what the reviewer is asking of us with this comment. We are not claiming that CUP1 copy number is not important in the provision of copper tolerance. In fact, the QTL analysis of F1 progeny from a cross between Cutol:CUP1high x Cusen:CUP1low strains demonstrated that high CUP1 copy number is a key feature of copper tolerance in wine yeast. This finding is consistent with the work of others and demonstrates that the copper tolerance phenotypes being investigated here are not specific to the condition being tested.

The subsequent QTL analysis, of a cross between Cutol:CUP1high x Cusen:CUP1high demonstrates the contribution of genetic features not previously associated with copper tolerance (SSU1 translocation).

To clarify this section we have reworded the paragraph describing the QTL analysis of F1 progeny from a cross between Cutol:CUP1high x Cusen:CUP1low strains as follows;

“An analysis of the SNP frequency in F1 progeny from AWRI 3001 (Cutol:CUP1high x Cusen:CUP1low) was undertaken. A clear divergence from the 1:1 expected parental genotypic ratios across the sensitive and resistant bulk pools was observed at base pair 210,000 on Chr VIII. This position corresponds almost exactly to the location of CUP1-1 and CUP1-2 (Error! Reference source not found.A). The major association on Chr VIII was consistent with the mean difference in CUP1 copy number between the two strains (Δ copy number = 23.1 copies [95CI, 19.9, 26.3]). This data explains the copper sensitivity of the AWRI 1537 strain and supports previous observations [18,19] that CUP1 is a key determinant of copper tolerance in yeast.”

Reviewer answer:

I apologize for the confusion as my comment was based on an incorrect interpretation, yet I believe the clarification added to the section is still valuable.

Author response: At least half of this section contains a discussion of genes that are down-regulated in this contrast, including commentary on genes related to thiamine metabolism and cell wall structure. While we agree with the reviewer that there is a lot of information in this data set that is not specifically presented (ie. we have left out any discussion of comparisons of low and high copper conditions), we have deliberately attempted to be concise with this section to leave space for the presentation of information relating to understanding the causes of SSU1 suppression of copper tolerance in CUP1 carrying yeast.

Reviewer answer:

Indeed, interesting down-regulated cases were commented on. My comment was focused on many genes under sulfate/ sulfur GO terms but as they are no longer present in the new analysis, my concern has dissipated.

Reviewer: L347 % of variance explained would be interesting

Author Response:

Apologies to the reviewer in this point but we do not understand what they are referring to with this comment

Reviewer answer:

(Now line 398 in the revised manuscript). I Apologies to the authors for the lack of detail in my comment. More generally, I suggest that adding the variance explained information from the ANOVA, in addition to the p-value, when commenting on the ANOVA results would be beneficial and provide more insight into the importance of some factors.

Reviewer: Fig 7 is lacking the strain AWRI 3471 as control.

Author response:

We did not include AWRI 3471 in this experiment. We had already shown that AWRI 3471 is unaffected by 10 mg/L of copper when growing in medium containing limited SO4 (Fig S7). To highlight this point we have added the following sentence to the manuscript; “In contrast, the growth of AWRI 3471 in SO4 limited medium was largely unaffected by 10mg/L copper and sugar utilisation was only slightly delayed (S7 Fig).”

Reviewer response:

I understand now why AWRI 3471 was not included in this experiment. In this case, as suggested by another reviewer, Supp Fig 7 would maybe deserve to be present as a main figure, and maybe instead of the actual Fig 7

I reiterate this possibility as I believe that the primary finding in this section is the difference in phenotypic response between the two strains.

Reviewer #2: The authors have satisfactorily answered all my comments.

Reviewer #3: I think the authors did an excellent job responding to my comments and those of the other reviewers. Their revision of the manuscript clarified the copper stress and conditions they are using. Altogether, the authors did a great job at reporting the relationship between sulphate and copper tolerance, and I recommend the publication of this nice paper.

**Have all data underlying the figures and results presented in the manuscript been provided?**

Reviewer #1: Yes

Reviewer #2: None

Reviewer #3: Yes

PLOS authors have the option to publish the peer review history of their article (what does this mean?). If published, this will include your full peer review and any attached files.

Reviewer #1: **Yes: **Emilien Peltier

Reviewer #2: **Yes: **Gilles Fischer

Reviewer #3: No

**Data Deposition**

http://datadryad.org/submit?journalID=pgenetics&manu=PGENETICS-D-22-01269R1

**Press Queries**

---

## [Editor Report · Acceptance letter]

24 Mar 2023

PGENETICS-D-22-01269R1 

SO_2_ and copper tolerance exhibit an evolutionary trade-off in *Saccharomyces cerevisiae*. 

Dear Dr Schmidt, 

We are pleased to inform you that your manuscript entitled "SO_2_ and copper tolerance exhibit an evolutionary trade-off in *Saccharomyces cerevisiae*." has been formally accepted for publication in PLOS Genetics! Your manuscript is now with our production department and you will be notified of the publication date in due course.

With kind regards,

Bernadett Koltai

PLOS Genetics

On behalf of:
